# Identification of fallopian tube microbiota and its association with ovarian cancer

Bo Yu[1,2]*, Congzhou Liu[3], Sean C Proll[3], Enna Manhardt[4], Shuying Liang[4], Sujatha Srinivasan[3], Elizabeth Swisher[4], David N Fredricks[3]*

[1]Department of Obstetrics and Gynecology, Stanford, United States; [2]Stanford Maternal & Child Health Research Institute, Stanford University School of Medicine, Stanford, United States; [3]Vaccine and Infectious Disease Division, Fred Hutchinson Cancer Center, Seattle, United States; [4]Department of Obstetrics and Gynecology, University of Washington, Seattle, United States

**\*For correspondence:**
byu1@stanford.edu (BY);
dfredric@fredhutch.org (DNF)

**Abstract** Investigating the human fallopian tube (FT) microbiota has significant implications for understanding the pathogenesis of ovarian cancer (OC). In this large prospective study, we collected swabs intraoperatively from the FT and other surgical sites as controls to profile the microbiota in the FT and to assess its relationship with OC. Eighty-one OC and 106 non-cancer patients were enrolled and 1001 swabs were processed for 16S rRNA gene PCR and sequencing. We identified 84 bacterial species that may represent the FT microbiota and found a clear shift in the microbiota of the OC patients when compared to the non-cancer patients. Of the top 20 species that were most prevalent in the FT of OC patients, 60% were bacteria that predominantly reside in the gastro-intestinal tract, while 30% normally reside in the mouth. Serous carcinoma had higher prevalence of almost all 84 FT bacterial species compared to the other OC subtypes. The clear shift in the FT microbiota in OC patients establishes the scientific foundation for future investigation into the role of these bacteria in the pathogenesis of OC.

## eLife assessment

Little is known about the role of the microbiome alterations in epithelial ovarian cancer. This **important** and rigorous study carefully examined the microbiome composition of 1001 samples from close to 200 ovarian cancer cases and controls, and presents **compelling** evidence that the fallopian tube microbiota are perturbed in ovarian cancer patients. These insights are expected to fuel further exploration into translational opportunities stemming from these findings.

## Introduction

The American Cancer Society estimates that in 2023, about 19,710 new cases of ovarian cancer (OC) will be diagnosed and about 13,270 individuals will die from OC in the United States (*Cancer Facts & Figures 2022, 2022*). Unlike most other cancers, the mortality rate for OC has declined only slightly in the last 40 years due to lack of early diagnostic tests and effective treatments. The 5-year survival rate for all types of OC averages 49.7%, much lower than many other cancers (*Cancer Stat Facts: Ovarian Cancer, 2019*). The high mortality rate of OC is linked to a lack of understanding of ovarian carcinogenesis and precursor lesions, contributing to the failure of early detection and late-stage diagnosis. Even the cell origins of various OC subtypes have not been fully appreciated until recently, with ovarian carcinoma representing a heterogeneous collection of cancer subtypes, each with distinct origins and molecular drivers (*Shih et al., 2021*). The fallopian tube (FT) epithelium is the likely origin of most high-grade serous carcinomas, the most common subtype of ovarian carcinoma. There is a

critical need for innovative research to illuminate the origins and pathogenesis of this deadly cancer, thus enabling targeted approaches for early detection, treatment, and prevention.

Based on the epidemiological data, we propose a novel hypothesis for ovarian carcinogenesis: ascending infection with some genital tract bacteria leads to inflammation in the FTs and DNA damage to cells contributing to neoplastic transformation. The evidence supporting this hypothesis includes: (1) The FTs form a conduit between the lower genital tract and the pelvic cavity (*Heller et al., 1996*; *Henderson et al., 1979*). (2) Blocking the communication between the FTs and the environment, such as through tubal ligation or hysterectomy, results in lower OC incidences (*Gaitskell et al., 2016*; *Wang et al., 2016*; *Parker et al., 2009*). (3) Increased cellular and bacterial transit between the lower genital tract and the peritoneal cavity, as in endometriosis or pelvic inflammatory disease, is associated with increased OC risks (*Melin et al., 2011*; *Lin et al., 2011*). (4) Inhibiting ovulation through oral contraceptives may decrease OC risks (*Narod et al., 1998*; *Tworoger et al., 2007*; *Ness et al., 2001*) by reducing the opportunities of incorporating ascending pathogens into the ovarian surface epithelium during ovulation-induced microtrauma. (5) The fallopian tubal epithelium has been implicated in recent studies as the site of origin for at least a substantial proportion of high-grade serous carcinomas (*Norquist et al., 2010*; *Medeiros et al., 2006*; *Lamb et al., 2006*). Furthermore, recent studies have demonstrated the presence of genital tract bacteria in the upper reproductive tract of women without known infections (*Mitchell et al., 2015*; *Chambers et al., 2021*; *Walther-António et al., 2016*; *Ventolini et al., 2022*). Therefore, bacteria ascending from the lower genital tract may reside in the FTs and could induce a pro-inflammatory environment, which could influence neoplasia of the tubal or ovarian epithelium, with outcome dependent on species of bacteria, concentration, host factors, and duration of exposure.

Even though the above epidemiological studies have hinted to the possible connection between an altered microbiota in the FT and OC, few studies have examined this plausible hypothesis. Even the existence of a microbiota in the FT has not been convincingly demonstrated, in part due to the technical challenges of accessing this site in a sterile fashion. Several studies have provided evidence that the FT may not be sterile (*Miles et al., 2017*; *Brewster et al., 2022*; *Asangba et al., 2023*), but rigorous controls and large sample sizes are needed to assess the microbiota in low-biomass samples. Here, we examine the FT microbiota from a large study population with collection of swabs in a sterile fashion in the operating room from both patients with OC and those without. The FT bacterial concentrations and compositions were compared between two groups: women who are cancer free, and women who have any type of OC. We built in numerous controls in our study design to help differentiate likely contaminants from signal of an FT microbiome.

## Results

### Patient overview

A total of 187 patients who met the inclusion and exclusion criteria (see Methods) were enrolled in this study. The patient characteristics were different between the cancer and non-cancer patients in their age and menopausal status as summarized in *Table 1*, with a higher percentage of the OC patients in the older and postmenopausal groups. Most OCs were diagnosed in stages III and IV, with high-grade serous carcinoma being the most common histology subtype (*Table 1*). Laparotomy was more commonly performed in OC patients (*Table 1*).

### Sample overview and low-biomass study design

Eight hundred and fifty-four biological samples and 147 non-biological laboratory controls were analyzed for microbiota composition (*Supplementary file 1*). Three hundred and sixty-nine swabs were collected from the FT and ovarian surfaces. We also designed additional layers of controls and collected from multiple sites for comparisons. To assess the microbial environment in the operating room, 130 'air swabs' were collected from the patient's room. To assess the cervical microbiota, 152 cervical swabs were collected before surgical area preparation and vaginal sterilization. To assess the skin microbiota that may be introduced into the pelvic cavity during laparoscopic port insertion, 81 'laparoscopic port swabs' were collected after the port insertion and before the start of operation. To assess the microbiota in the abdominal cavity away from the surgical or cancer sites, 122 'paracolic gutter swabs' were collected from the right paracolic gutter before the start of operation.

**Table 1.** Demographic characteristics of participants.

| | Ovarian cancer patients (*n* = 81) | Non-cancer patients (*n* = 106) | p-value |
|---|---|---|---|
| Average age at diagnosis | 59.7 | 55.3 | 0.052 |
| Average age at surgery | 59.6 | 51.6 | <0.001 |
| Post-menopausal | 62 (76.5%) | 48 (45.3%) | <0.001 |
| Race | | | 0.492 |
| - Asian | 7 (8.6%) | 4 (3.8%) | |
| - Black | 1 (1.2%) | 3 (2.8%) | |
| - Other | 6 (7.4%) | 5 (4.7%) | |
| - White | 64 (79.0%) | 92 (86.80%) | |
| - Undisclosed | 3 (3.7%) | 2 (1.9%) | |
| Ethnicity | | | 0.218 |
| - Hispanic | 3 (3.7%) | 1 (0.9%) | |
| - Non-Hispanic | 73 (90.1%) | 102 (96.2%) | |
| - Undisclosed | 5 (6.2%) | 3 (2.8%) | |
| Cancer stage at surgery | | | N/A |
| 1 | 19 (23.5%) | - | |
| 2 | 6 (7.45%) | - | |
| 3 | 39 (48.2%) | - | |
| 4 | 16 (19.8%) | - | |
| NA | 1 (1.2%) | - | |
| Cancer grade | | | N/A |
| 1 | 6 (7.4%) | - | |
| 2 | 3 (3.7%) | - | |
| 3 | 64 (79.0%) | - | |
| 4 | 1 (1.2%) | - | |
| Tumor histology type | | | N/A |
| - Adenocarcinoma | 3 | - | |
| - Carcinosarcoma | 1 | - | |
| - Clear cell | 8 | - | |
| - Endometrioid | 10 | - | |
| - Granulosa cell | 1 | - | |
| - Mucinous | 2 | - | |
| - Serous | 53 | - | |
| - Transitional cell | 1 | - | |
| - Borderline serous* | - | 8 | |
| - Other | - | 98 | |
| Surgical type | | | <0.001 |
| - Laparotomy | 75 (92.6%) | 28 (26.4%) | |
| - Laparoscopy | 3 (3.7%) | 42 (39.6%) | |
| - Robotic assisted | 3 (3.7%) | 36 (34.0%) | |

*Table 1 continued on next page*

*Table 1 continued*

|  | Ovarian cancer patients (*n* = 81) | Non-cancer patients (*n* = 106) | p-value |
|---|---|---|---|
| Pelvic washing |  |  | <0.001 |
| - Positive | 43 (53.1%) | 3 (2.8%) |  |
| - Negative | 22 (27.2%) | 99 (93.4%) |  |
| - NA | 16 (19.8%) | 4 (3.8%) |  |

*Borderline serous tumors are considered in the non-cancer category in this study due to the non-invasiveness of these tumors.

To assess potential contamination of DNA extraction reagents, during each batch of DNA extractions we included 'buffer controls' that contained only DNA extraction buffer without any swab or other DNA. These 36 'buffer controls' were processed and then sequenced together with the swab samples. To assess potential contamination during PCR and sequencing, we also included multiple no-template PCR controls that contained only PCR reagents and water without extracted DNA or buffer (*n* = 111 total) during each sequencing run (*Supplementary file 1*).

## Bacterial concentration

The bacterial concentrations of FT and ovarian surfaces averaged 2.5 copies of 16S rRNA genes/ µl of DNA (standard deviation, 4.6). Compared to the buffer controls and the OR air swabs, the FT and ovarian surface swabs contained higher concentrations of bacterial DNA than the controls (p-value <0.001) (*Figure 1*; *Supplementary file 2*). As expected, the cervical swabs contained thousands of times higher bacterial concentrations than the FT and ovarian surface swabs (p-value <0.001) (*Figure 1*). The bacterial concentrations from the paracolic gutter were similar to those from the FT and ovarian surface swabs (p-value = 0.11) (*Figure 1*; *Supplementary file 2*).

## FT microbiota analysis

In the raw sequencing analysis, 892 bacterial species were present with at least 100 reads in at least one sample. After a series of filtering as described in Methods, 84 bacterial species were present in at least one cervical and one FT sample with over 100 reads while not present in non-biological and air swabs (*Supplementary file 3*). These 84 bacterial species may represent the FT microbiota that are present in some women. When analyzing the 340 samples that contained at least one read of these 84 species, the Principle Component Analysis (PCA) plot showed a high level of similarity between the fallopian tube/ovarian surface and the paracolic gutter swabs, indicating these species represent the shared microbiota community in the abdominal and pelvic cavity within each individual (*Figure 2A*). An overview of these 84 bacterial species and their presence in each FT/ovarian surface sample shows that some of the bacterial species are prevalent in multiple individuals, such as *Klebsiella* or *Anaerococcus*, while others are only present in a few individuals, such as *Casaltella* or *BV-associated bacterium 2* (*BVAB2*) (*Figure 3*). Clustering did not identify any obvious groupings by processing batch, cancer status, menopausal status, or age for these putative FT bacterial species (*Figure 3*). The Shannon index was calculated for each sample to determine if there were differences in bacterial diversity. There were 726 bacterial species present in the cervix, FT/ovarian surface, and paracolic gutter after filtering out the bacteria present in no-template controls and DNA extraction controls (i.e. after step #5 in *Supplementary file 3*). The cervical samples had lower Shannon diversity index than the FT/ovarian surface and paracolic gutter samples, while the non-cancer versus OC samples did not show any statistically significant difference in Shannon diversity within each sample type other than the cervical samples (*Figure 2B*).

## Comparison between OC and non-cancer microbiota

We compared the prevalence of each of these 84 bacterial species in patients with or without OC (*Figure 4*). Some bacteria, such as *Streptococcus parasanguinis* or *Neisseriaceae*, are more common in FT/ovarian surface samples from OC patients, while others (e.g. *Ruminiclostridium*, *Dialister invisus*, or *Bacteroides dorei*) are only present in OC patients albeit overall representation was low. After ranking the FT bacteria based on the prevalence difference, we found a clear shift in the microbiota of

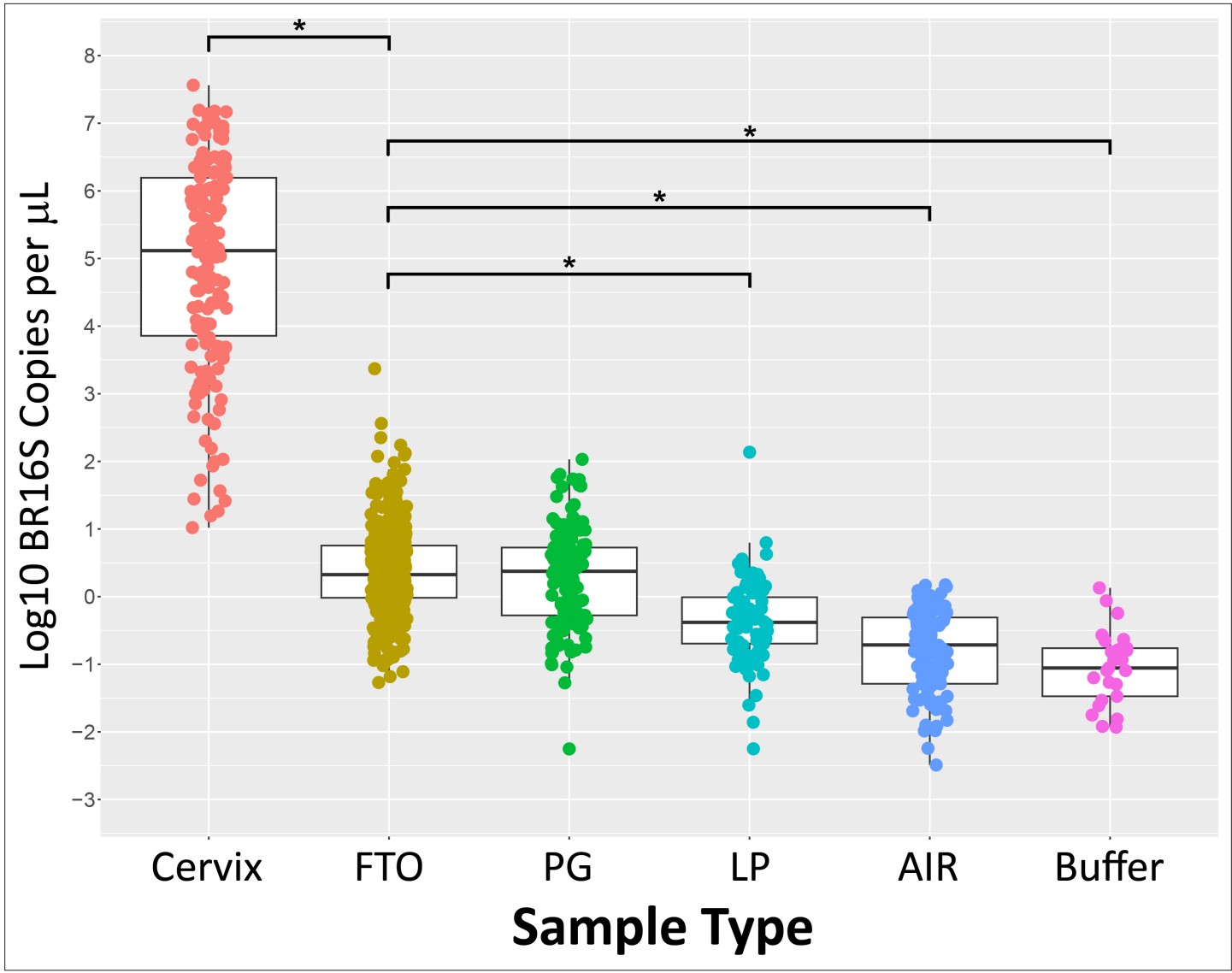

**Figure 1.** Bacterial concentration of each swab sample and DNA extraction controls as measured by broad-range 16S rRNA gene PCR. *p-value <0.001, paired *t*-test, comparing each sample type with fallopian tube/ovarian surface (FTO) samples. PG = paracolic gutter; LP = laparoscopic port.

the OC patients when compared to the non-cancer patients. Interestingly, 90% of the top 20 species that were most prevalent in the FT of OC patients were bacteria that predominantly exist outside the female reproductive tract. Among these, 12 (60%) predominantly reside in the gastrointestinal tract, such as *Klebsiella*, *Faecalibacterium prausnitzii*, *Ruminiclostridium*, and *Roseburia*. Six of the top 20 species (or 30%) normally reside in the mouth, such as *Streptococcus mitis*, *Corynebacterium simulans/striatum*, and *D. invisus*. On the contrary, vaginal bacterial species, such as *Corynebacterium amycolatum*, *Gardnerella*, and *Lactobacillus iners*, are more prevalent in the FT from non-cancer patients, representing 75% of the top 20 bacterial species that are most prevalent in non-cancer patients. This pattern was also consistently observed in the paracolic gutter samples which share the same low-biomass environment of abdominal and pelvic cavity , suggesting that the above findings of FT microbiota were not random. When the histology subtypes were considered in the analysis, we found that the most common subtype, serous carcinoma, had higher prevalence of almost all 84 FT bacterial species compared to the other OC subtypes (*Figure 5*), indicating a higher level of perturbance in the FT microbiota in the serous carcinoma cases.

Since the majority of laparotomy cases were performed in OC patients and the non-OC cases were mostly performed with laparoscopy or robotic-assisted laparoscopy, we evaluated the influence

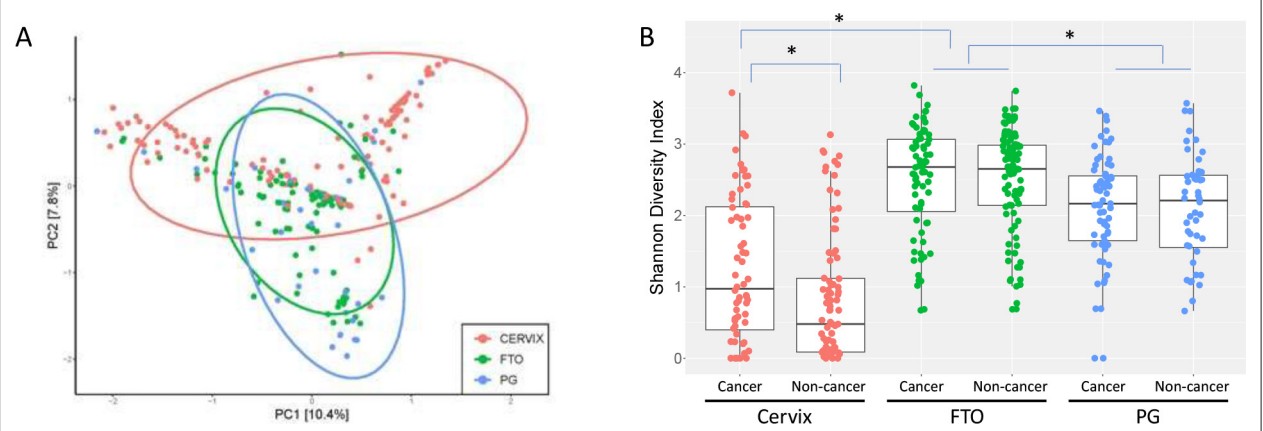

**Figure 2.** Diversity of identified bacterial species. (**A**) PCA plot of candidate fallopian tube (FT) microbiota (84 bacterial species). FT (green) samples have more similarities with paracolic gutter (blue) than with cervical (red) samples. (**B**) Shannon diversity plot of 726 bacterial species identified after filtering step #5 in **Supplementary file 3**. *p-value <0.001, paired *t*-test, comparing between sample types and between cancer versus non-cancer samples within the same sample type. FTO = fallopian tube/ovarian surface; PG = paracolic gutter.

of the surgical type on the FT microbiota profiles. When only laparotomy cases were considered in the comparison of OC versus non-cancer patients, the pattern of prevalence of majority of the 84 FT bacterial species was nearly identical as the overall comparison including all surgical types (**Supplementary file 4**). When laparotomy cases were compared to laparoscopic or robotic cases within the non-cancer patients, the prevalence of most of the 84 FT bacterial species was similar between the surgical types (**Supplementary file 5**). These analyses demonstrate that the differences we observed in the putative FT microbiota between OC and non-OC patients were not caused by the differences in surgical approaches.

## Discussion

In this large prospective study, we analyzed the microbiota on swabs collected in a sterile fashion in the operating room during surgery, including from the FT and ovarian surface in 187 patients with or without OC, with the objective of determining an FT microbiome and identifying microbiome differences associated with OC. After detailed analyses and filtering out likely contaminants, we found 84 bacterial species that are present in FT or ovarian surface, which may represent an FT microbiota in some women. Comparison between the OC and non-cancer cases revealed a significantly higher prevalence of bacterial species that predominantly do not reside in the reproductive tract in patients with OC, while the FT from non-cancer patients favored vaginal bacterial species. It is unclear whether OC leads to this microbiota shift or if the microbiota shift precedes OC.

We proposed a novel hypothesis of ovarian carcinogenesis based on epidemiological evidence before initiating our study, namely that ascending infection with some genital tract bacteria leads to inflammation in the FT and causes DNA damage to cells resulting in OC. Our study showed the existence of bacterial species at a low concentration in the FT and ovarian surface in most women without overt signs of infection. Some of these bacterial species overlapped between the FT and the cervical swabs, indicating potential ascension from the lower to the upper genital tract. Some other bacterial species are not typical vaginal microbiota, and the origins of these bacteria in the FT samples are less clear. Patients with OC, especially the serous carcinoma subtype, showed significantly higher prevalence of many of the bacterial species we identified as the putative FT microbiota, indicating a potential link between a perturbed FT microbiota and OC. The FT microbiota may be different in different OC histology subtype, which was consistent with a recent study with limited sample size (**Asangba et al., 2023**).

One of the most interesting findings from our study is that most of the top bacterial species that were more prevalent in OC patients had predominant niche outside of female reproductive tract, normally residing in the gastrointestinal or oral tract. Some of these bacterial species were also identified in previous small studies examining the bacteria incorporated into the FT and OC tissues, such

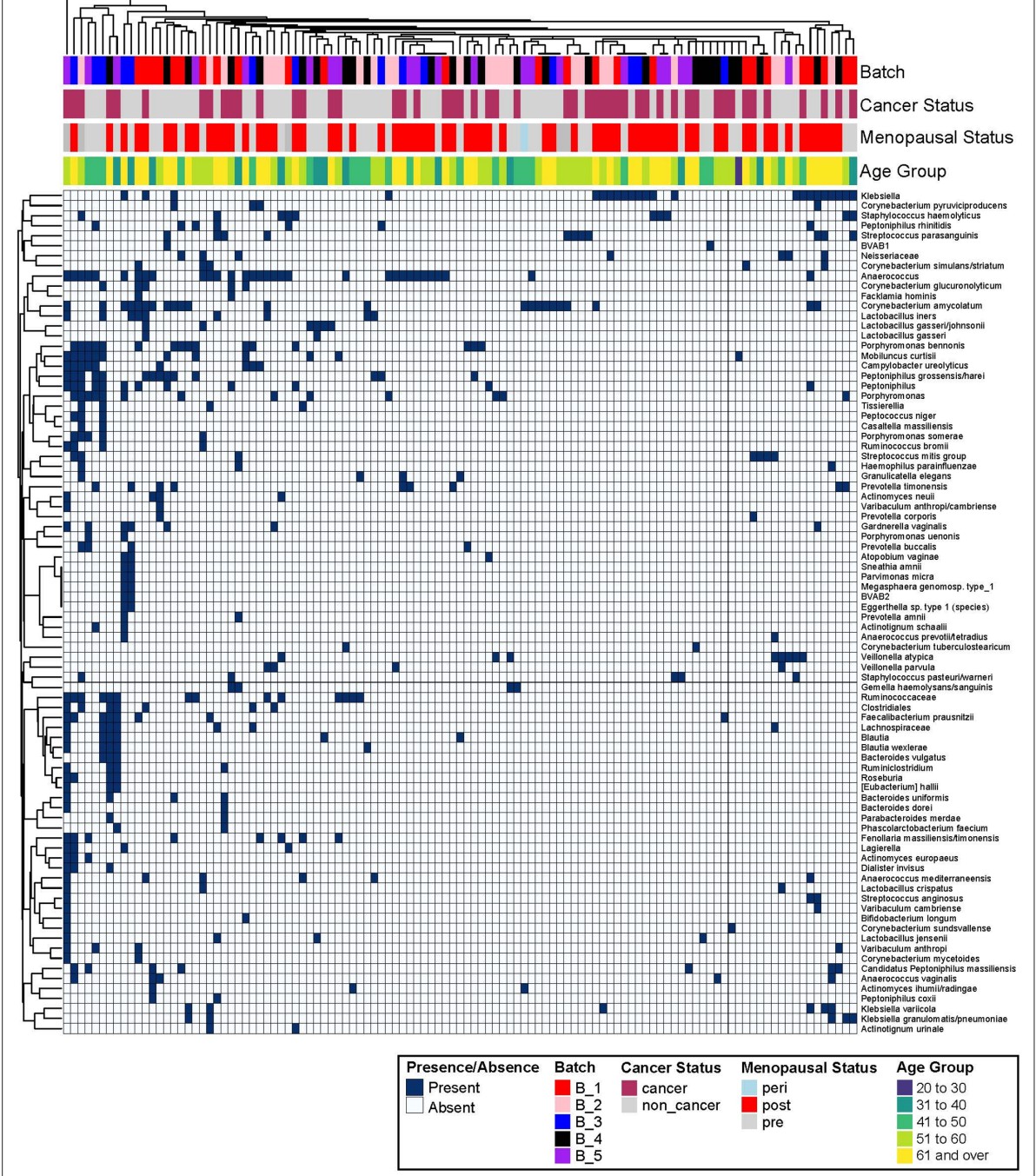

**Figure 3.** Overview of all participants with processing batch, cancer status, menopausal status, and age in relation to fallopian tube (FT) microbiome taxa. Each column is a patient, and each row is a bacterial species. The top rows indicate the metadata of each sample as denoted by the color coding of batch, cancer status, menopausal status, and age group.

as *Anaerococcus*, *Klebsiella*, *Bacteroides*, and *Streptococcus* (*Asangba et al., 2023*; *Banerjee et al., 2017*). The origin and route of ascension of these bacteria to the FT and ovarian surfaces remain unknown. In late-stage cancers, even when there are no overt signs of gastrointestinal invasion, the permeability of the intestinal tract may be increased which can increase the prevalence of these bacterial species in the FT, ovarian surfaces, and paracolic gutters. The causal effect of bacterial or other pathogens on the FT inflammation or carcinogenesis will need to be investigated in future studies

| | Fallopian Tube and Ovary | | Cervix | | Paracolic Gutter | | Predominant Niche |
|---|---|---|---|---|---|---|---|
| | Cancer | Non-cancer | Cancer | Non-cancer | Cancer | Non-cancer | |
| *Klebsiella* | 16.05 | 9.43 | 1.23 | 1.89 | 17.28 | 7.55 | G |
| *Faecalibacterium prausnitzii* | 7.41 | 1.89 | 7.41 | 3.77 | 2.47 | 0.00 | G |
| *Ruminiclostridium* | 4.94 | 0.00 | 4.94 | 0.94 | 3.70 | 0.00 | G |
| *Roseburia* | 4.94 | 0.00 | 2.47 | 2.83 | 2.47 | 0.00 | G |
| *Clostridiales* | 6.17 | 1.89 | 3.70 | 0.94 | 4.94 | 0.00 | B |
| *Streptococcus mitis* group | 6.17 | 1.89 | 2.47 | 0.00 | 1.23 | 0.00 | O |
| *Corynebacterium simulans/striatum* | 4.94 | 0.94 | 2.47 | 0.94 | 1.23 | 0.00 | O |
| *[Eubacterium] hallii* | 3.70 | 0.00 | 2.47 | 1.89 | 0.00 | 0.94 | G |
| *Dialister invisus* | 3.70 | 0.00 | 1.23 | 0.94 | 2.47 | 0.00 | O |
| *Fenollaria massiliensis/timonensis* | 6.17 | 2.83 | 13.58 | 7.55 | 2.47 | 0.00 | GV |
| *Anaerococcus mediterraneensis* | 4.94 | 1.89 | 19.75 | 9.43 | 1.23 | 0.00 | V |
| *Blautia* | 4.94 | 1.89 | 1.23 | 1.89 | 1.23 | 0.94 | G |
| *Bacteroides uniformis* | 3.70 | 0.94 | 3.70 | 1.89 | 2.47 | 0.94 | G |
| *Gemella haemolysans/sanguinis* | 3.70 | 0.94 | 2.47 | 2.83 | 2.47 | 0.00 | O |
| *Ruminococcus bromii* | 3.70 | 0.94 | 1.23 | 1.89 | 3.70 | 0.00 | G |
| *Tissierella* | 3.70 | 0.94 | 2.47 | 0.94 | 1.23 | 0.00 | G |
| *Granulicatella elegans* | 3.70 | 0.94 | 3.70 | 0.94 | 0.00 | 0.00 | O |
| *Streptococcus parasanguinis* | 7.41 | 4.72 | 2.47 | 0.94 | 0.00 | 0.00 | O |
| *Bacteroides dorei* | 2.47 | 0.00 | 2.47 | 0.94 | 1.23 | 0.00 | G |
| *Parabacteroides merdae* | 2.47 | 0.00 | 2.47 | 0.00 | 2.47 | 0.00 | G |
| *Actinotignum urinale* | 2.47 | 0.00 | 3.70 | 0.94 | 0.00 | 0.00 | V |
| *Phascolarctobacterium faecium* | 2.47 | 0.00 | 1.23 | 0.94 | 1.23 | 0.00 | G |
| *Corynebacterium sundsvallense* | 2.47 | 0.00 | 2.47 | 0.00 | 0.00 | 0.00 | B |
| *Neisseriaceae* | 4.94 | 2.83 | 1.23 | 0.94 | 0.00 | 0.00 | OV |
| *Porphyromonas somerae* | 3.70 | 1.89 | 6.17 | 0.94 | 4.94 | 0.00 | VS |
| *Blautia wexlerae* | 3.70 | 1.89 | 3.70 | 0.94 | 1.23 | 0.94 | G |
| *Staphylococcus pasteuri/warneri* | 3.70 | 1.89 | 1.23 | 0.94 | 1.23 | 0.00 | S |
| *Peptoniphilus* | 7.41 | 5.66 | 16.05 | 12.26 | 2.47 | 0.00 | GV |
| *Lactobacillus crispatus* | 2.47 | 0.94 | 13.58 | 26.42 | 3.70 | 0.94 | V |
| *Bacteroides vulgatus* | 2.47 | 0.94 | 3.70 | 2.83 | 2.47 | 0.94 | G |
| *Haemophilus parainfluenzae* | 2.47 | 0.94 | 6.17 | 0.94 | 1.23 | 0.94 | O |
| *Peptococcus niger* | 2.47 | 0.94 | 4.94 | 1.89 | 2.47 | 0.00 | GO |
| *Actinomyces europaeus* | 2.47 | 0.94 | 2.47 | 1.89 | 1.23 | 0.00 | GO |
| *Lactobacillus gasseri/johnsonii* | 3.70 | 2.83 | 13.58 | 13.21 | 2.47 | 1.89 | GV |
| *Klebsiella variicola* | 3.70 | 2.83 | 0.00 | 1.89 | 2.47 | 4.72 | G |
| *Varibaculum anthropi* | 2.47 | 1.89 | 19.75 | 4.72 | 0.00 | 0.00 | V |
| *Lagierella* | 2.47 | 1.89 | 6.17 | 1.89 | 2.47 | 0.00 | G |
| *Varibaculum cambriense* | 1.23 | 0.94 | 13.58 | 2.83 | 0.00 | 0.00 | V |
| *Facklamia hominis* | 1.23 | 0.94 | 11.11 | 1.89 | 1.23 | 0.00 | V |
| *Bifidobacterium longum* | 1.23 | 0.94 | 7.41 | 2.83 | 0.00 | 0.00 | G |
| *Casaltella massiliensis* | 1.23 | 0.94 | 4.94 | 0.94 | 3.70 | 0.00 | G |
| *Prevotella corporis* | 1.23 | 0.94 | 4.94 | 2.83 | 0.00 | 0.00 | GO |
| *Corynebacterium tuberculostearicum* | 1.23 | 0.94 | 6.17 | 0.94 | 0.00 | 0.00 | S |
| *Prevotella amnii* | 1.23 | 0.94 | 1.23 | 2.83 | 0.00 | 0.94 | V |
| *Varibaculum anthropi/cambriense* | 1.23 | 0.94 | 4.94 | 0.94 | 0.00 | 0.00 | V |
| *Lactobacillus gasseri* | 1.23 | 0.94 | 0.00 | 2.83 | 0.00 | 1.89 | V |
| *Corynebacterium mycetoides* | 1.23 | 0.94 | 2.47 | 0.00 | 0.00 | 0.00 | V |
| *Campylobacter ureolyticus* | 4.94 | 4.72 | 16.05 | 8.49 | 3.70 | 0.94 | G |
| *Candidatus Peptoniphilus massiliensis* | 3.70 | 3.77 | 6.17 | 0.94 | 4.94 | 1.89 | G |
| *Lachnospiraceae* | 3.70 | 3.77 | 4.94 | 3.77 | 1.23 | 0.00 | B |
| *Ruminococcaceae* | 7.41 | 7.55 | 7.41 | 0.00 | 6.17 | 1.89 | G |
| *Anaerococcus vaginalis* | 2.47 | 2.83 | 20.99 | 12.26 | 3.70 | 0.94 | V |
| *Klebsiella granulomatis/pneumoniae* | 2.47 | 2.83 | 0.00 | 1.89 | 4.94 | 3.77 | G |
| *Corynebacterium glucuronolyticum* | 2.47 | 2.83 | 1.23 | 0.94 | 2.47 | 0.94 | V |
| *Streptococcus anginosus* | 1.23 | 1.89 | 30.86 | 14.15 | 1.23 | 0.00 | O |
| *Atopobium vaginae* | 1.23 | 1.89 | 4.94 | 12.26 | 0.00 | 0.94 | V |
| *Staphylococcus haemolyticus* | 4.94 | 5.66 | 1.23 | 0.94 | 1.23 | 0.94 | S |
| *Lactobacillus jensenii* | 1.23 | 2.83 | 14.81 | 22.64 | 0.00 | 0.00 | V |
| *Prevotella buccalis* | 1.23 | 2.83 | 12.35 | 2.83 | 3.70 | 0.94 | OV |
| *Actinomyces neuii* | 1.23 | 2.83 | 8.64 | 4.72 | 1.23 | 0.94 | V |
| *Corynebacterium pyruviciproducens* | 1.23 | 2.83 | 8.64 | 2.83 | 0.00 | 0.00 | V |
| *Veillonella parvula* | 1.23 | 2.83 | 1.23 | 0.94 | 0.00 | 0.00 | OV |
| *Porphyromonas uenonis* | 0.00 | 1.89 | 9.88 | 10.38 | 1.23 | 0.94 | V |
| *Peptoniphilus coxii* | 0.00 | 1.89 | 8.64 | 10.38 | 0.00 | 0.94 | OV |
| *Actinotignum schaalii* | 0.00 | 1.89 | 11.11 | 6.60 | 1.23 | 0.00 | V |
| *Anaerococcus prevotii/tetradius* | 0.00 | 1.89 | 6.17 | 7.55 | 0.00 | 0.00 | V |
| *Sneathia amnii* | 0.00 | 1.89 | 2.47 | 6.60 | 0.00 | 0.94 | V |
| *Megasphaera* genomosp. type_1 | 0.00 | 1.89 | 2.47 | 5.66 | 1.23 | 0.94 | V |
| *Eggerthella* sp. type 1 (species) | 0.00 | 1.89 | 1.23 | 7.55 | 0.00 | 0.00 | V |
| *Parvimonas micra* | 0.00 | 1.89 | 2.47 | 5.66 | 1.23 | 0.00 | V |
| BVAB2 | 0.00 | 1.89 | 1.23 | 3.77 | 0.00 | 0.94 | V |
| BVAB1 | 0.00 | 1.89 | 1.23 | 1.89 | 0.00 | 0.00 | V |
| *Prevotella timonensis* | 3.70 | 5.66 | 18.52 | 10.38 | 1.23 | 0.00 | V |
| *Peptoniphilus grossensis/harei* | 6.17 | 8.49 | 30.86 | 17.92 | 3.70 | 2.83 | G |
| *Mobiluncus curtisii* | 4.94 | 7.55 | 20.99 | 8.49 | 3.70 | 1.89 | V |
| *Actinomyces ihumii/radingae* | 0.00 | 2.83 | 2.47 | 0.94 | 1.23 | 1.89 | G |
| *Veillonella atypica* | 2.47 | 5.66 | 2.47 | 1.89 | 0.00 | 0.94 | GO |
| *Anaerococcus* | 14.81 | 18.87 | 11.11 | 6.60 | 9.88 | 2.83 | GOV |
| *Lactobacillus iners* | 2.47 | 6.60 | 18.52 | 23.58 | 1.23 | 1.89 | V |
| *Gardnerella vaginalis* | 1.23 | 5.66 | 14.81 | 17.92 | 1.23 | 1.89 | V |
| *Porphyromonas* | 3.70 | 9.43 | 3.70 | 0.94 | 2.47 | 0.94 | GOV |
| *Porphyromonas bennonis* | 4.94 | 11.32 | 13.58 | 6.60 | 6.17 | 0.00 | V |
| *Peptoniphilus rhinitidis* | 0.00 | 6.60 | 3.70 | 0.00 | 1.23 | 0.00 | O |
| *Corynebacterium amycolatum* | 4.94 | 17.92 | 8.64 | 1.89 | 7.41 | 6.60 | V |

**Predominant Niche**

- Gastrointestinal
- Vaginal
- Oral
- Skin
- Broadly present or multiple niches

**Figure 4.** Prevalence of the 84 bacterial species in ovarian cancer versus non-cancer cases. Each number is the percentage of individuals in each category with the presence of each bacterial species. G = gastrointestinal; O = oral; V = vaginal; S = skin; B = broadly present.

The online version of this article includes the following source data for figure 4:

**Source data 1.** Prevalence of the 84 bacterial species in ovarian cancer versus non-cancer cases.

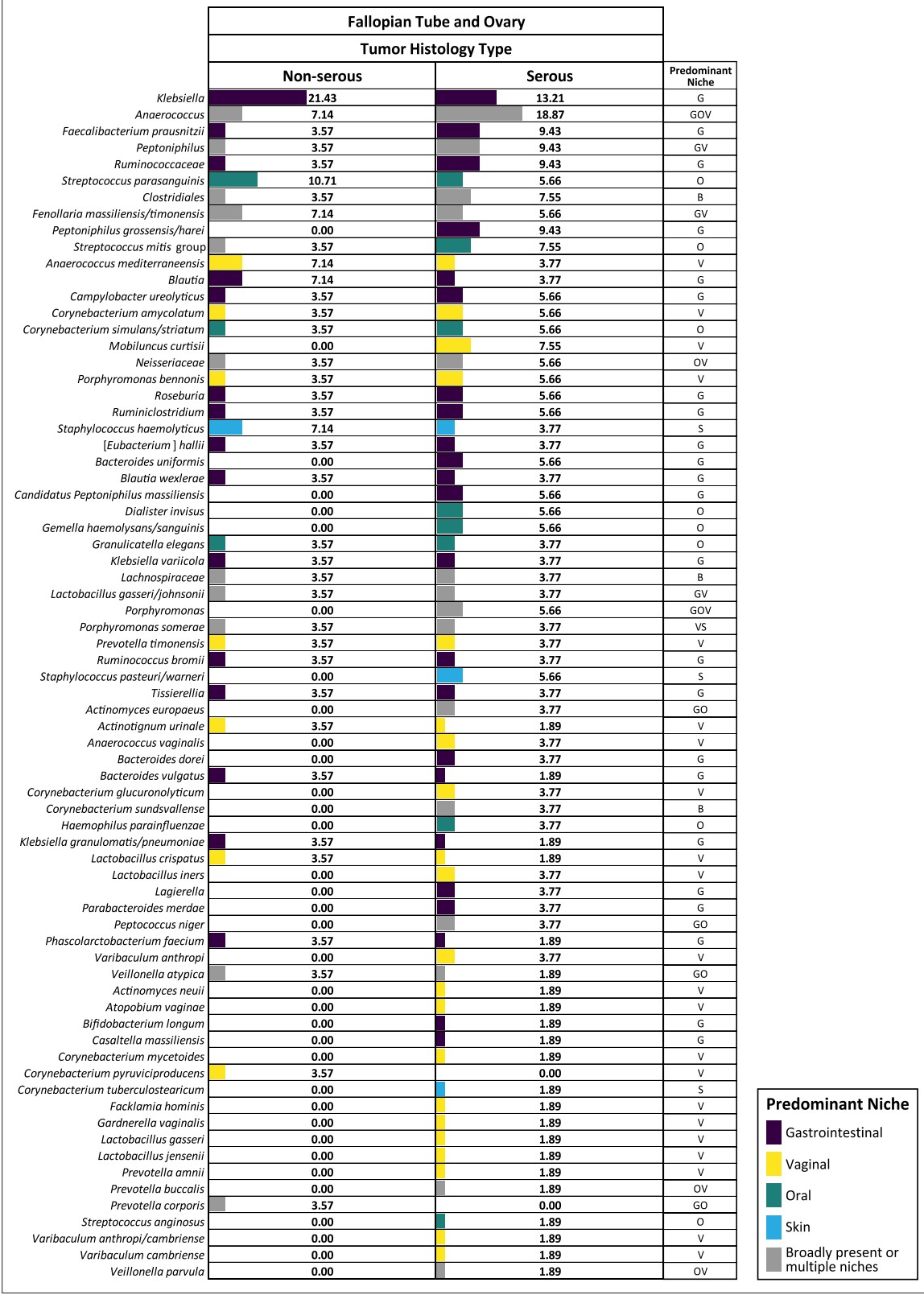

**Figure 5.** The bacterial prevalence in fallopian tube (FT) samples from ovarian cancer patients by histology subtypes (non-serous versus serous carcinoma). Each number is the percentage of individuals in each category with the presence of each bacterial species.

The online version of this article includes the following source data for figure 5:

**Source data 1.** Prevalence in ovarian cancer patients by histology subtypes.

using preclinical models. However, our study provides a list of top bacterial candidates that can be further investigated.

Our approach in collecting swabs in a sterile fashion in the operating room before the surgical case starts is unique and has not been used in previous studies (*Miles et al., 2017*; *Brewster et al., 2022*; *Asangba et al., 2023*). This approach enables us to evaluate the microbiota in each individual patient at multiple sites to infer transfer among sites. Previous studies mostly used the tissues collected from the pathology department after gross examination, which can introduce contamination. These studies infrequently included negative controls to assess for environmental or laboratory contamination, which is critically important when assessing a low-biomass microbiome (*Salter et al., 2014*). We collected swabs not only from the site of interest, that is, FT and ovarian surface, but also from key control sites linked to cases, such as the air in the OR, the laparoscopic port, the paracolic gutter, and the cervix. These biological controls, along with the DNA extraction and PCR controls during the sample processing and sequencing processes, provide a fuller picture of the patient-specific microbiota and differentiate the biologically meaningful bacterial species from likely contaminants. The approach taken here may serve as a guide for others undertaking microbiome studies of low-biomass environments, particularly when considering study design, types of controls, and analytical approaches (*O'Callaghan et al., 2020*). The sample size is among the largest in microbiome studies of low-biomass sites, which is helpful in differentiating signal from the noise in this setting. All previous microbiome studies that included FT as a study site had less than 40 patients in each group (*Miles et al., 2017*; *Brewster et al., 2022*; *Asangba et al., 2023*).

There are some limitations to our study. Due to the cross-sectional nature of the study, we have limited ability to link specific bacteria to ovarian carcinogenesis, as we would need to demonstrate that exposure to bacteria precedes the cancer. However, identifying associations between FT microbiota and OC is a critical first step. Further investigations, especially backed by in vitro studies, are needed to test our initial hypotheses. Due to the large sample size, we processed the samples in five batches and sequenced the libraries on four plates. Even though we separated the cervical samples with the low-biomass samples on different sequencing plates to prevent cross-contamination, we sequenced the low-biomass samples on three different sequencing plates, which could introduce bias due to batch effects. However, all samples were processed by a single experienced technician, and each sequencing plate contained samples from different processing batches, which helped to diminish potential batch effects. By requiring the bacterial species be present in both FT/ovarian surface swabs and the cervical swabs, we may have missed part of the FT microbiota that may have originated from sources other than the lower genital tract.

In conclusion, in this large prospective study with extensive controls, we analyzed the FT swabs from 187 patients together with controls that were collected intraoperatively in a sterile fashion, and identified a putative FT microbiota that exists in many women without any overt signs of infection. Patients with OC had a high prevalence of some key bacterial species, indicating a potentially perturbed microbiota co-existing with OC. Further studies are needed to investigate whether there is a causal relationship between these bacterial species and ovarian carcinogenesis.

## Materials and methods

### Study population

Specimens and clinical information were obtained from patients who provided informed consent under a protocol approved by the institutional review board at University of Washington (Protocol # 2872). Surgical indications included OCs of various histological types, risk-reducing salpingo-oophorectomies due to germline *BRCA* or other mutations, and benign gynecological disorders such as ovarian cysts or endometriosis. Exclusion criteria include pelvic inflammatory disease, presence of an intrauterine device, use of antibiotics, endometrial biopsy, intrauterine device removal, or hysteroscopy in the 30 days prior to the intended enrollment.

### Sample collection

During the surgeries that involve salpingectomies, we collected FT swabs in the operating room in a sterile fashion. From most enrolled patients, we collected from the following sites: (1) a swab from the air in the operating room, (2) a cervical swab before vaginal sterile prep, (3) a swab from the

laparoscopy port after the port insertion and before introduction of any instrument in laparoscopic cases, (4) a swab from the unaffected peritoneum in the paracolic gutter above the liver, and (5) a swab from the FT and ovarian surface on each side. In some patients, the swabs may not have been collected from all sites due to surgical accessibility, time constraint, or unintentional omission. All biological samples were assigned anonymous participant identification numbers, which were matched with participants' names in a securely stored link file.

## DNA extraction and quantitative PCR

All swabs were vortexed and washed in 200 µl 0.9% saline, in which saline was filtered by MilliporeSigma Amicon Ultra Centrifugal 100 kDa Filter Units (Thermo Fisher Scientific, Waltham, MA). DNA from swab prep mixture was extracted using the QIAamp BiOstic Bacteremia DNA KIT (QIAGEN, Germantown, MD) and eluted in a mixture of 25 µl EB buffer provided in the kit plus 25 µl 0.2× Tris–Ethylenediaminetetraacetic acid(EDTA). In each quality control quantitative polymerase chain reaction (qPCR) assay, 5 µl DNA was loaded per reaction. Digestion buffer control was DNA extracted from sham swabs to assess presence of contaminant in kit or during extraction procedure. Absence of PCR inhibitor was confirmed by an internal amplification control (IAC) qPCR (*Khot et al., 2008*). IAC qPCR was designed to compare amplification of spiked-in jellyfish gDNA between DNA samples and water. Total bacterial concentration was measured by a broad-range qPCR targeting V3–V4 region of the bacterial 16S rRNA gene (*Srinivasan et al., 2012*).

## Broad-range PCR and deep sequencing of 16S rRNA gene amplicons

Broad-range amplicon PCR targeting V3–V4 hypervariable region of the 16S rRNA gene was performed using an adapter attached 338 F and 806 R primer formulation (*Srinivasan et al., 2012*; *Golob et al., 2017*). The optimal quantity of DNA per reaction is 2.4e+6 copies of the 16S rRNA gene. Maximum allowed volume was added in amplicon PCR (20 µl DNA/30 µl reaction) for samples from collection sites other than the cervix, which were defined as lower biomass samples. Filtered water was added to PCRs as a no-template (negative) control. A sham DNA extraction was performed to assess contamination of extraction regents and the resulting DNA added to PCRs as another form of negative control with each batch of samples processed. Amplicon PCR products were cleaned using Agencourt AMPure XP beads (Beckman Coulter, Indianapolis, IN) to remove primer dimer and reaction buffers. Cleaned amplicons were applied to barcoded Index PCR using NexteraXT index kits v2 (Illumina, San Diego, CA). Per index PCR, 15 µl of lower biomass amplicon or 5 µl of higher biomass amplicon was loaded. Index PCR product went through a second-round bead clean and was eluted in 30 µl 1× Tris-EDTA (TE) buffer. DNA concentration of each sample was measured by the Quant-iT dsDNA assay kit-HS (Thermo Fisher Scientific, Waltham, MA). For the low-biomass samples, the entire volume of cleaned index PCR product was pooled into sub pools, which then were concentrated roughly 50 times by MilliporeSigma Amicon Ultra Centrifugal 10 kDa Filter Units (Thermo Fisher Scientific, Waltham, MA). Equal quantities (8 nM or lower) of subpools were pooled into a master pool, which was then subjected to deep sequencing.

Next-generation sequencing (NGS) of the bacterial 16S rRNA gene PCR product was performed on the Illumina MiSeq instrument (Illumina). PhiX Control Library v3 (Illumina) was combined with the amplicon library at 15% for high biomass samples and at 20% for low-biomass samples.

## NGS data analysis

Sequence reads were processed using the DADA2 package (*Callahan et al., 2016*) for error correction, dereplication, paired-end assembly, and chimera removal and a list of unique sequence variants (SVs) were generated. A custom vaginal reference set was used to assign taxonomy to individual SVs (*Srinivasan et al., 2012*). Taxonomic assignments were made as described previously (*Srinivasan et al., 2021*). Briefly, a multiple sequence alignment of both query and reference sequences was created using *cmalign* (*Nawrocki and Eddy, 2013*) and query sequences were placed on the phylogenetic tree using *pplacer* (*Matsen et al., 2010*). Taxonomy was assigned to each unique SV based on location on the tree. Bacterial taxa represented by fewer than 100 reads in a sample were excluded from that sample to minimize environmental contaminant sequences from being included in the final dataset. The samples from the same site and the same patient were merged as one sample for downstream analysis. The NGS data were filtered using a processing pipeline that we developed in this

study to track samples and sequence types (*Supplementary file 3*). R packages were used for the statistical analyses and graphics. Sequences have been submitted to the NCBI Short Read Archive (accession number PRJNA975142).

## Acknowledgements

The authors acknowledge the surgeons and staff in the Department of Obstetrics and Gynecology at University of Washington for collecting samples for this study, and Noah Hoffman, MD, PhD from the Department of Laboratory Medicine at the University of Washington for his assistance with taxonomic analysis of the data.

## Additional information

### Funding

| Funder | Grant reference number | Author |
|---|---|---|
| National Institutes of Health | K08CA222835 | Bo Yu |
| Stanford Maternal and Child Health Research Institute | Akiko Yamazaki and Jerry Yang Faculty Scholar Fund in Pediatric Translational Medicine | Bo Yu |
| Seattle Translational Tumor Research | | Elizabeth Swisher |
| National Institutes of Health | R01AI139189 | Bo Yu David N Fredricks |
| Stanford Cancer Institute | SCI Women's Cancer Center Innovation Award | Bo Yu |

The funders had no role in study design, data collection, and interpretation, or the decision to submit the work for publication.

### Author contributions

Bo Yu, Conceptualization, Data curation, Formal analysis, Supervision, Funding acquisition, Writing – original draft, Writing – review and editing; Congzhou Liu, Enna Manhardt, Shuying Liang, Data curation; Sean C Proll, Formal analysis, Writing – review and editing; Sujatha Srinivasan, Formal analysis; Elizabeth Swisher, David N Fredricks, Conceptualization, Supervision, Funding acquisition, Methodology, Writing – review and editing

### Author ORCIDs

Bo Yu ⬤ https://orcid.org/0000-0002-6051-077X

### Ethics

Specimens and clinical information were obtained from patients who provided informed consent under a protocol approved by the Institutional Review Board at University of Washington (Protocol # 2872).

Reviewer #2 (Public Review): https://doi.org/10.7554/eLife.89830.3.sa1
Reviewer #3 (Public Review): https://doi.org/10.7554/eLife.89830.3.sa2
Author Response https://doi.org/10.7554/eLife.89830.3.sa3

## Additional files

### Supplementary files

• Supplementary file 1. Summary of samples sequenced.

• Supplementary file 2. Bacterial concentration (log$_{10}$[16S rRNA genes/µl of DNA]) of each sample type and the p-value of each comparison.

• Supplementary file 3. Processing steps and result summary after each step.

• Supplementary file 4. Comparison of overall and laparotomy cases in the bacterial prevalence in fallopian tube (FT) samples from ovarian cancer versus non-cancer patients. Each number is the percentage of individuals in each category with the presence of each bacterial species.

• Supplementary file 5. The bacterial prevalence in fallopian tube (FT) samples from non-cancer patients: comparison of laparoscopic/robotic and laparotomy cases. Each number is the percentage of individuals in each category with the presence of each bacterial species.

• MDAR checklist

## Data availability

Sequencing data have been deposited in NCBI BioProject under the accession code PRJNA975142.

The following dataset was generated:

| Author(s) | Year | Dataset title | Dataset URL | Database and Identifier |
| --- | --- | --- | --- | --- |
| Fredricks D | 2023 | Identification of fallopian tube microbiota and its association with ovarian cancer | https://www.ncbi.nlm.nih.gov/bioproject/PRJNA975142 | NCBI BioProject, PRJNA975142 |

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
