## [Editor Report · eLife assessment]

Little is known about the role of the microbiome alterations in epithelial ovarian cancer. This **important** and rigorous study carefully examined the microbiome composition of 1001 samples from close to 200 ovarian cancer cases and controls, and presents **compelling** evidence that the fallopian tube microbiota are perturbed in ovarian cancer patients. These insights are expected to fuel further exploration into translational opportunities stemming from these findings.

---

## [Referee Report · Reviewer #2 (Public Review)]

The authors aimed to investigate the microbiota present in the fallopian tubes (FT) and its potential association with ovarian cancer (OC). They collected swabs intraoperatively from the FT and other surgical sites as controls to profile the FT microbiota and assess its relationship with OC.

They observed a clear shift in the FT microbiota of OC patients compared to non-cancer patients. Specifically, the FT of OC patients had more types of bacteria typically found in the gastrointestinal tract and the mouth. In contrast, vaginal bacterial species were more prevalent in non-cancer patients. Serous carcinoma, the most common OC subtype, showed a higher prevalence of almost all FT bacterial species compared to other OC subtypes.

The strengths of the study include its large sample size, rigorous collection methods, and use of controls to identify the possible contaminants. Additionally, the study employed advanced sequencing techniques for microbiota analysis. However, there are some weaknesses to consider. The study relied on swabs collected intraoperatively, which may not fully represent the microbiota in the FT during normal physiological conditions. The study also did not establish causality between the identified bacteria and OC but rather demonstrated an association. Regardless, the findings are important and these questions need to be addressed by future studies. A few additions in data representation and analysis are instead recommended.

Overall, the authors achieved their aims of identifying the FT microbiota and assessing its relationship with OC. The results support the conclusion that there is a clear shift in the FT microbiota in OC patients, paving the way for further investigations into the role of these bacteria in the pathogenesis of ovarian cancer.

The identification of specific bacterial species associated with OC could contribute to the development of novel diagnostic and therapeutic approaches. The study design and the data generated here can be valuable to the research community studying the microbiota and its impact on cancer development. However, further research is needed to validate these findings and elucidate the underlying mechanisms linking the FT microbiota shift and OC.

---

## [Referee Report · Reviewer #3 (Public Review)]

The findings of Bo Yu and colleagues titled "Identification of fallopian tube microbiota and its association with ovarian cancer: a prospective study of intraoperative swab collections from 187 patients" describes the identification of the fallopian tube microbiome and relationship with ovarian cancer. The studies are highly rigorous obtaining specimens from the fallopian tube, ovarian surfaces, paracolic gutter of patients of known or suspected ovarian cancer or benign tumor patients. The investigators took great care to insure there was no or limited contamination including test the surgical suite air, as the test locations are from low abundance microbiota. The findings provide evidence that the microbiota in the fallopian tube, especially in ovarian cancer has similarities to gut microbial communities. This is a potentially novel observation.

The studies investigate the microbiome of >1000 swabs from 81 ovarian cancer and 106 non-cancer patients. The sites collected are low biomass microbiota making the study particularly challenging. The studies provide descriptive evidence that the ovarian cancer fallopian tube microbiota contain species that are similar to the gut microbiota. In contrast the fallopian tube microbiota of non-cancer patients that exhibit more similarity to the uterine/cervical microbiota. This may be a relevant observation but is highly descriptive with limited insights on the functional relevance.

The data indicate the presence of low biomass FT microbiota. The findings support the existence of FT microbiota in ovarian cancer that appears to be related to gut microbial species. While interesting, there is no insights on how and why these microbial species are found in the FT. The studies only identify the species but there is no transcriptomic analysis to provide an indication on whether the bacteria are activating DNA damage pathways. This is an interesting observation that requires more insights to address how these bacteria reach the fallopian tube and a related question is whether these bacteria are found in the peritoneum.

An additional concern is whether these data can be used to develop biomarkers of disease and early detection of disease.

---

## [Author Response]

**Reviewer #1 (Public Review):**
The authors propose a hypothesis for ovarian carcinogenesis based on epidemiological data, and more specifically they suggest that the latter relates to ascending genital tract "infection" or "dysbiosis", the resulting fallopian tube inflammation ultimately predisposing to ovarian cancer.While this hypothesis would ideally be addressed in a longitudinal set-up with repeated female genital tract sampling, such an approach is obviously hard to realize. Rather, the authors present this hypothesis as a rationale for a cross-sectional study involving 81 patients with ovarian cancer (most with the most common subtype of high grade serous ovarian carcinoma, though other subtypes were also included), as well as 106 control patients with various non-infectious conditions including endometriosis and benign ovarian cysts. In all patients was there a comprehensive microbiome sampling of ovarian surface/fallopian tube, cervix and peritoneal cavity as well sampling of a number of potential sources of contamination, including surgery sites, ambient environment, consumables used in the DNA extraction and sequencing pipeline, etc. In line with the hypothesis presented at the outset, species with a threshold of at least 100 reads in both at least one cervical and at least one fallopian tube sample, while absent from environmental swabs, were considered relevant to the postulated pathway.Remarkably, fallopian tube microbiota in ovarian cancer patients tended to cluster more closely to those retrieved from the paracolic gutter, than fallopian tube microbiota in non-cancer controls, which showed more relative similarity to vaginal/genital tract microbiota.Although not really addressed by the authors, there also seem to be quite a few differences, at least in terms of abundance, in cervical microbiota between ovarian cancer patients and controls as well, which is an interesting finding, even when accounting for differences in age distribution between ovarian cancer patients and included control patients.Overall, very few data are available thus far on the upper genital tract/fallopian tube microbiome, while also invariably controversial, as it has proven extremely difficult to obtain pelvic samples in a valid, "sterile" manner, i.e. without affecting a resident low-biomass microbiome to be analyzed. The authors took a number of measures to counter so, and in this respect, this is likely the largest and most valid study on the subject, even though biases and contamination can never be completely excluded in this context.As such, I believe the strength of this study and paper primarily relates to the rigour of the methodology, thereby giving us a valuable insight in the presumed fallopian tube/ovarian surface microbiome, which may definitely serve as an impetus and a reference to future translational ovarian cancer research, or ovarian microbiome research for that matter.I believe that the authors should acknowledge in more detail, that the data obtained from their cross-sectional study, valid as these are, do not provide any direct support to the hypothesis - albeit also plausible - set forth, a discussion that I somehow missed to a certain extent. It is important to realize in this and related contexts that neoplasia may well induce microbiome alterations through a variety of mechanisms, hence microbiome alterations not per se being causative. Conclusions should therefore be more reserved. Along the same lines, potential biases introduced through the selection of control patients (some detail here would be insightful) also deserves some discussion, as it is not known, whether other conditions such as benign ovarian cysts or endometriosis have some relationship with the human microbiome, be it causative or 'reversely causative', see for instance very recent work in Science Translational Medicine.

We appreciate the reviewer’s detailed review and thoughtful comments. We have added the following sentences in the Discussion to address the reviewer’s concern: “Due to the cross-sectional nature of the study, we have limited ability to link specific bacteria to ovarian carcinogenesis, as we would need to demonstrate that exposure to bacteria precedes the cancer. However, identifying associations between FT microbiota and OC is a critical first step. Further investigations, especially backed by in vitro studies, are needed to test our initial hypotheses.”

**Reviewer #2 (Public Review):**
The authors aimed to investigate the microbiota present in the fallopian tubes (FT) and its potential association with ovarian cancer (OC). They collected swabs intraoperatively from the FT and other surgical sites as controls to profile the FT microbiota and assess its relationship with OC.They observed a clear shift in the FT microbiota of OC patients compared to non-cancer patients. Specifically, the FT of OC patients had more types of bacteria typically found in the gastrointestinal tract and the mouth. In contrast, vaginal bacterial species were more prevalent in non-cancer patients. Serous carcinoma, the most common OC subtype, showed a higher prevalence of almost all FT bacterial species compared to other OC subtypes.The strengths of the study include its large sample size, rigorous collection methods, and use of controls to identify the possible contaminants. Additionally, the study employed advanced sequencing techniques for microbiota analysis. However, there are some weaknesses to consider. The study relied on swabs collected intraoperatively, which may not fully represent the microbiota in the FT during normal physiological conditions. The study also did not establish causality between the identified bacteria and OC but rather demonstrated an association. Regardless, the findings are important and these questions need to be addressed by future studies. A few additions in data representation and analysis are instead recommended.Overall, the authors achieved their aims of identifying the FT microbiota and assessing its relationship with OC. The results support the conclusion that there is a clear shift in the FT microbiota in OC patients, paving the way for further investigations into the role of these bacteria in the pathogenesis of ovarian cancer.The identification of specific bacterial species associated with OC could contribute to the development of novel diagnostic and therapeutic approaches. The study design and the data generated here can be valuable to the research community studying the microbiota and its impact on cancer development. However, further research is needed to validate these findings and elucidate the underlying mechanisms linking the FT microbiota shift and OC.

We appreciate the reviewer’s detailed review and positive comments.

**Reviewer #3 (Public Review):**
The findings of Bo Yu and colleagues titled "Identification of fallopian tube microbiota and its association with ovarian cancer: a prospective study of intraoperative swab collections from 187 patients" describes the identification of the fallopian tube microbiome and relationship with ovarian cancer. The studies are highly rigorous obtaining specimens from the fallopian tube, ovarian surfaces, paracolic gutter of patients of known or suspected ovarian cancer or benign tumor patients. The investigators took great care to ensure there was no or limited contamination including test the surgical suite air, as the test locations are from low abundance microbiota. The findings provide evidence that the microbiota in the fallopian tube, especially in ovarian cancer has similarities to gut microbial communities. This is a potentially novel observation.The studies investigate the microbiome of >1000 swabs from 81 ovarian cancer and 106 non-cancer patients. The sites collected are low biomass microbiota making the study particularly challenging. The studies provide descriptive evidence that the ovarian cancer fallopian tube microbiota contain species that are similar to the gut microbiota. In contrast the fallopian tube microbiota of non-cancer patients that exhibit more similarity to the uterine/cervical microbiota. This may be a relevant observation but is highly descriptive with limited insights on the functional relevance.The data indicate the presence of low biomass FT microbiota. The findings support the existence of FT microbiota in ovarian cancer that appears to be related to gut microbial species. While interesting, there is no insights on how and why these microbial species are found in the FT. The studies only identify the species but there is no transcriptomic analysis to provide an indication on whether the bacteria are activating DNA damage pathways. This is an interesting observation that requires more insights to address how these bacteria reach the fallopian tube and a related question is whether these bacteria are found in the peritoneum.An additional concern is whether these data can be used to develop biomarkers of disease and early detection of disease. can the investigators detect the ovarian cancer FT microbiota in cervical/vaginal secretions? That may yield more significant insights for the field.

We appreciate the reviewer’s detailed review and thoughtful comments. We have added the following sentences in the Discussion to acknowledge the reviewer’s concern: “Due to the cross-sectional nature of the study, we have limited ability to link specific bacteria to ovarian carcinogenesis, as we would need to demonstrate that exposure to bacteria precedes the cancer. However, identifying associations between FT microbiota and OC is a critical first step. Further investigations, especially backed by in vitro studies, are needed to test our initial hypotheses.”

**Reviewer #1 (Recommendations For The Authors):**

I have no additional comments here.

**Reviewer #2 (Recommendations For The Authors):**
The data analysis and data representation could be improved by the following points:1. To compare the microbiota and assess the overall microbiota structure difference between the cancer vs non cancer cohort alpha- and beta-diversity of the microbial communities can be conducted.1. A differential abundance analysis could also be conducted to assess the differences at the genera and taxa level between the cancer vs non cancer cohorts.1. The analysis suggested above can also be conducted in the serous vs non serous cancer cohorts.1. In Figure 4 and 5 it would be more intuitive to show the predominant niche of each bacterium by color coding

We appreciate these helpful suggestions from the reviewer. We have added Figure 2B to address the diversity as well as the differences between cancer versus non-cancer cohorts. We have added in the Results section the description of our findings in Figure 2B. We have added color coding to Figure 4 and 5 as the reviewer suggested.

**Reviewer #3 (Recommendations For The Authors):**
These studies are interesting but are very descriptive with no obvious approaches for understanding the mechanisms of FT microbiota in ovarian cancer. The identification of these bacteria is not sufficient to draw implications on their impact on ovarian cancer development or progression. This needs to be addressed.

We agree with the reviewer and have added the following sentences in the Discussion to acknowledge the reviewer’s concern: “Due to the cross-sectional nature of the study, we have limited ability to link specific bacteria to ovarian carcinogenesis, as we would need to demonstrate that exposure to bacteria precedes the cancer. However, identifying associations between FT microbiota and OC is a critical first step. Further investigations, especially backed by in vitro studies, are needed to test our initial hypotheses.”